# Rethinking the Fourier-Mellin Transform: Multiple Depths in the Camera's View

Qingwen Xu [1,2,3,*], Haofei Kuang [1], Laurent Kneip [1] and Sören Schwertfeger [1]

1 School of Information Science Technology, ShanghaiTech University, Shanghai 201210, China; kuanghf@shanghaitech.edu.cn (H.K.); lkneip@shanghaitech.edu.cn (L.K.); soerensch@shanghaitech.edu.cn (S.S.)
2 Chinese Academy of Sciences, Shanghai Institute of Microsystem & Information Technology, Shanghai 200050, China
3 University of Chinese Academy of Sciences, Beijing 100049, China
* Correspondence: xuqw@shanghaitech.edu.cn

**Abstract:** Remote sensing and robotics often rely on visual odometry (VO) for localization. Many standard approaches for VO use feature detection. However, these methods will meet challenges if the environments are feature-deprived or highly repetitive. Fourier-Mellin Transform (FMT) is an alternative VO approach that has been shown to show superior performance in these scenarios and is often used in remote sensing. One limitation of FMT is that it requires an environment that is equidistant to the camera, i.e., single-depth. To extend the applications of FMT to multi-depth environments, this paper presents the extended Fourier-Mellin Transform (eFMT), which maintains the advantages of FMT with respect to feature-deprived scenarios. To show the robustness and accuracy of eFMT, we implement an eFMT-based visual odometry framework and test it in toy examples and a large-scale drone dataset. All these experiments are performed on data collected in challenging scenarios, such as, trees, wooden boards and featureless roofs. The results show that eFMT performs better than FMT in the multi-depth settings. Moreover, eFMT also outperforms state-of-the-art VO algorithms, such as ORB-SLAM3, SVO and DSO, in our experiments.

**Keywords:** drone-based remote sensing; Fourier-Mellin transform; visual odometry; 3D perception; spectral registration

## 1. Introduction

Visual odometry (VO) plays an important role in remote sensing and robotics [1–3], as many applications rely on visual information, especially in GNSS-denied environments. In general, VO estimates the 6DoF (Degrees of Freedom) motion of the camera in 3D space. Popular examples of such algorithms are ORB-SLAM [4], SVO [5], LSD-SLAM [6] and DSO [7]. Certain applications of VO only estimate 4DoF, while avoiding any roll or pitch of the camera. Examples for those are down-looking cameras on satellites [8], aerial vehicles [9,10] or underwater vehicles [11].

Usually, Fourier-Mellin transform (FMT) is used to estimate such 4DoF motion for remote sensing [12–14]. FMT is based on Fourier transform analysis, which is important for image analysis [15,16]. It was used to estimate the motion between two images with the phase-only matched filter [17]. Reddy and Chatterji [18] presented the classic Fourier-Mellin transform to calculate the rotation, zoom and translation between images. *Please note that the zoom is described as scaling in [18], which represents the image change when the camera moves along the direction perpendicular to the imaging plane, but we use "zoom" in this paper to distinguish it from "re-scaling" of visual odometry. Monocular cameras cannot recover absolute scale, thus all translations among frames are re-scaled w.r.t the estimated translation between the first two frames [1].* In [19,20], FMT was improved to speed up the computation and boost the robustness. Also, FMT was shown to be more accurate and faster than SIFT

in certain environments in [19]. In addition, the visual odometry based on FMT performs more accurate and robust than that based on different features, such as ORB and AKAZE, especially in feature-deprived environments [21]. Most of current VO methods rely on features or pixel brightness. For example, ORB-SLAM uses ORB feature detectors to find correspondences between two images; SVO, LSD-SLAM and DSO estimate the motion between two frames based on the brightness consistency. There are also some methods using global appearance descriptors, which are more robust in feature-deprived scenarios. FMT is one of them. Furthermore, [22,23] compare the performance of different holistic descriptors for localization and mapping, such as discrete Fourier transform, principal components analysis and histogram of oriented gradients.

Due to FMT's robustness and high accuracy, it has been successfully applied in multiple applications, such as image registration [11,24,25], fingerprint image hashing [26], visual homing [27], point cloud registration [28], 3D modeling [29], remote sensing [12,30], and localization and mapping [31,32]. However, it requires that the capture device doesn't roll or pitch and that the environment is planar and parallel to the imaging plane. There are already several efforts on solving the first restriction. For instance, Lucchese calculated the affine transform via optimization based on the affine FMT analysis [33]. In [34], the oversampling technology and Dirichlet-based phase filter were used to make FMT robust to some image skew. Moreover, the sub-image extraction strategy [21,35–37] is popular in addressing the 3D motion problem. In this paper, we mainly focus on the second case, i.e., to relax the constraints of equidistance and planar environments. If the depths of objects are different, the pixels' motion will be different when the camera's motions are the same, which is due to perspective projection. Since FMT can only gives the image motion of the dominant depth, the camera's speed cannot be correctly inferred from the FMT's results when the dominant plane changes. Thus, an FMT-based visual odometry cannot work in multi-depth scenarios. For example, Figure 1 shows a multi-depth scenario, which contains buildings with different heights, lower ground and river. If images are collected in such scenarios with a down-looking camera mounted on an Unmanned Aerial Vehicle (UAV), FMT may first track the building roofs and then track the lower ground, such that it cannot estimate the camera's motion correctly because the dominant depth in the camera's view changes.

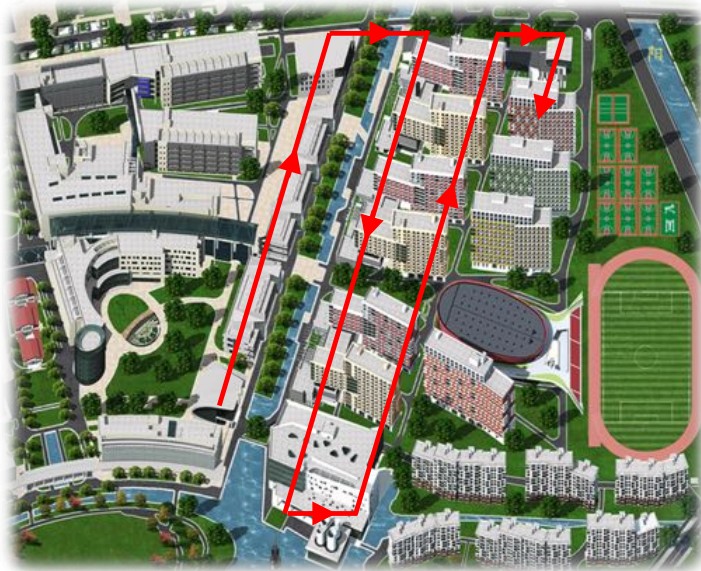

**Figure 1.** A UAV's flying trajectory over a campus. A down-looking camera is equipped on it.

To overcome the drawback of FMT, this paper presents the extended Fourier-Mellin Transform: eFMT. It extends FMT's 3D translation (translation and zoom), while keeping the original rotation estimation, because multiple depths result in multiple zooms and

translations, which will be discussed in detail in Section 3. Since FMT has already been used in all kinds of applications, such as remote sensing, image registration, localization and mapping, 3D modeling, visual homing, etc. (see above), we see a great potential of eFMT further enlarging the application scenarios of FMT. In this paper, we proceed to a highly practically relevant application of our proposed eFMT odometry algorithm, which is the motion estimation in the scenario of Figure 1 with a down-looking camera.

As we will also shown in this paper, in contrast to FMT, feature based and direct visual odometry frameworks usually do not perform well in challenging environments, such as low-texture surfaces (e.g., lawn, asphalt), underwater and fog scenarios [20]. The main advantage of FMT over other approaches—its robustness—is preserved in eFMT. To maximize the robustness and accuracy of FMT, we use the implementation of the improved FMT in [19,20] as comparison and build eFMT upon it in this paper. Our main contributions are summarized as follows:

- To the best of our knowledge, we are the first to apply the zoom and translation estimation in FMT to multi-depth environments. Our method is more general than FMT but maintains its strengths;
- We implement an eFMT-based visual odometry (VO) framework for one potential use-case of eFMT;
- We provide benchmarks in multi-depth environments between the proposed eFMT, the improved FMT [19,20], and popular VO approaches. The state-of-the-art VO methods, ORB-SLAM3 [4], SVO [5] and DSO [7], are chosen as comparison because they are the representative of feature-based, semi-direct and direct VO methods, respectively [38].

The rest of this paper is structured as follows: Section 2 recalls the classic FMT algorithm; Section 3 formulates the image registration in the multi-depth environment; Then we propose the eFMT algorithm for multi-depth visual odometry in Section 4; Experiments and analysis are present in Section 6; Finally, we conclude our work in Section 8.

## 2. Classical FMT

This section recaps the main idea of classic FMT [18]. Given two image signals $I_1, I_2$, the relationship between them is

$$
\begin{aligned}
I_2(x, y) = I_1(&zx \cos \theta_0 - zy \sin \theta_0 + x_0, \\
&zx \sin \theta_0 + zy \cos \theta_0 + y_0)
\end{aligned}
\tag{1}
$$

where $z$ and $\theta_0$ are constant and represent the zoom and rotation, respectively, and $(x_0, y_0)$ is the translation between $I_1$ and $I_2$. The motion parameters $(z, \theta, x_0, y_0)$ can be estimated by FMT via the following steps:

- Fourier transform on the image signals from both sides of Equation (1):

$$
\begin{aligned}
\mathcal{F}_2(\xi, \eta) = \quad & e^{-j2\pi(\xi x_0 + \eta y_0)} z^{-2} \\
& \mathcal{F}_1(z^{-1}\xi \cos \theta_0 - z^{-1}\eta \sin \theta_0, \\
& z^{-1}\xi \sin \theta_0 + z^{-1}\eta \cos \theta_0)
\end{aligned}
\tag{2}
$$

- Convert the magnitude $\mathcal{M}$ of Equation (2) in polar coordinates, ignoring the coefficients:

$$
\mathcal{M}_2(\rho, \theta) = \mathcal{M}_1(z^{-1}\rho, \theta - \theta_0) .
\tag{3}
$$

- Take the logarithm of $\rho$ of Equation (3):

$$
\mathcal{M}_2(\xi, \theta) = \mathcal{M}_1(\xi - d, \theta - \theta_0) ,
\tag{4}
$$

where $\xi = \log \rho, d = \log z$.

- Obtain $z$ and $\theta_0$ from Equation (4) based on the shift property of the Fourier Transform. Re-rotate and re-zoom $I_2$ to $I_2'$ so that

$$I_2'(x, y) = I_1(x - x_0, y - y_0) .$$ (5)

Accordingly,

$$\mathcal{F}'_2(\xi, \eta) = e^{-j2\pi(\xi x_0 + \eta y_0)} \mathcal{F}_1(\xi, \eta) .$$ (6)

Thus, all the motion parameters $(z, \theta_0, x_0, y_0)$ can be calculated by conducting phase correlation on Equations (4) and (5). Taking Equation (5) as an example, we first calculate the cross-power spectrum by

$$Q = \frac{\mathcal{F}_1(\xi, \eta) \circ \mathcal{F}_2'^*(\xi, \eta)}{|\mathcal{F}_1(\xi, \eta) \circ \mathcal{F}_2'^*(\xi, \eta)|} ,$$ (7)

where $\circ$ is the element-wise product and $^*$ represents the complex conjugate. By applying the inverse Fourier transform, we can obtain the normalized cross-correlation

$$q = \mathcal{F}^{-1}\{Q\} ,$$ (8)

which is also called *phase shift diagram (PSD)* in this paper. Then the translation $(x_0, y_0)$ corresponds to the location of the highest peak in $q$:

$$(x_0, y_0) = \arg\max_{(x,y)}\{q\} .$$ (9)

In the implementation the PSD is discretized into a grid of cells. Note that there exist partial non-corresponding regions between two frames due to the motion. Instead of contributing to the highest peak, these regions generate noise in the PSD. Since the energy of this noise is distributed over the PSD, it will not influence the detection and position of highest peak when the overlap between the frames is big enough.

Classical FMT describes the transformation between two images, which corresponds to the 4DoF motion of the camera, including 3DoF translation (zoom is caused by the translation perpendicular to the imaging plane) and yaw (assume $z$-axis is perpendicular to the imaging plane). However, as we mentioned in Section 1, it is limited to single-depth environments because it assumes zoom $z$ and translation $(x_0, y_0)$ as consistent and unique, which does not hold in multi-depth environments. In the next section, we formulate the image transformation in the multi-depth scenarios, i.e., considering multi-zoom and multi-translation. The solution provided by eFMT to handle this issue is presented in Section 4.

### 3. Problem Formulation

This section formulates the general image transformation with the 4DoF camera motion in multi-depth scenarios.

Given a pixel $p = [x, y]^\top$ of $I_1$, it is normalized to

$$\bar{p} = \begin{bmatrix} f_x & 0 & c_x \\ 0 & f_y & c_y \\ 0 & 0 & 1 \end{bmatrix}^{-1} \begin{bmatrix} x \\ y \\ 1 \end{bmatrix} = \begin{bmatrix} \frac{1}{f_x}(x - c_x) \\ \frac{1}{f_y}(y - c_y) \\ 1 \end{bmatrix}$$

with the focal length $f_x, f_y$ and image center $(c_x, c_y)$. Assume the pixel $p$ corresponds to the 3D point $P$ with depth $\delta$, then the coordinate of $P$ in the $I_1$'s frame is

$$P = \begin{bmatrix} \frac{\delta}{f_x}(x - c_x) \\ \frac{\delta}{f_y}(y - c_y) \\ \delta \end{bmatrix} .$$

Suppose the transformation between the camera poses of $I_1$ and $I_2$ is a 4DoF motion with the rotation around the camera principal axis, i.e., yaw $\theta$, the 2D translation in the imaging plane $(\Delta x, \Delta y)$, and the translation perpendicular to the imaging plane $\Delta \delta$, then $P$ in the $I_1$'s frame is projected to $I_2$ at point $p'$:

$$\begin{bmatrix} f_x & 0 & c_x \\ 0 & f_y & c_y \\ 0 & 0 & 1 \end{bmatrix} \begin{bmatrix} \cos\theta & -\sin\theta & 0 \\ \sin\theta & \cos\theta & 0 \\ 0 & 0 & 1 \end{bmatrix} P + \begin{bmatrix} \Delta x \\ \Delta y \\ \Delta\delta \end{bmatrix},$$

that is

$$p' = \begin{bmatrix} \frac{\delta}{\delta+\Delta\delta}(x\cos\theta - y\sin\theta) \\ +\frac{1}{\delta+\Delta\delta}(-\delta c_x\cos\theta + \delta c_y\sin\theta + f_x\Delta x) + c_x \\ \\ \frac{\delta}{\delta+\Delta\delta}(x\sin\theta + y\cos\theta) \\ +\frac{1}{\delta+\Delta\delta}(-\delta c_x\sin\theta - \delta c_y\cos\theta + f_y\Delta y) + c_y \end{bmatrix}.$$

Thus we can derive a general equation

$$\begin{aligned} I_2(x,y) &= I_1(z_\delta(x\cos\theta_0 - y\sin\theta_0) + x_\delta, \\ &\quad z_\delta(x\sin\theta_0 + y\cos\theta_0) + y_\delta) \end{aligned} \tag{10}$$

to describe the pixel transformation between $I_1$ and $I_2$, where $\theta_0 = \theta$,

$$z_\delta = \frac{\delta}{\delta + \Delta\delta}, \tag{11}$$

$$x_\delta = \frac{1}{\delta + \Delta\delta}(-\delta c_x\cos\theta + \delta c_y\sin\theta + f_x\Delta x) + c_x \tag{12}$$

and

$$y_\delta = \frac{1}{\delta + \Delta\delta}(-\delta c_x\sin\theta - \delta c_y\cos\theta + f_y\Delta y) + c_y . \tag{13}$$

It can be found that a zoom $z_\delta$ and a translation $(x_\delta, y_\delta)$ of a pixel depend on its depth $\delta$, while rotation $\theta_0$ is independent. Equation (1) is a simplification of Equation (10) under the condition that the depth $\delta$ of each pixel is the same. For $I_1$ and $I_2$ in a multi-depth scenario, there will be multiple solutions to Equations (10)–(13), depending on the depth of the individual pixel, so there are multiple zooms and translations. The energy of the cells in the PSD is positively correlated with the number of pixels with depth $\delta$ for which $(x_\delta, y_\delta)$ falls in that cell. Since FMT assumes an equidistant environment, the depth $\delta$ is considered constant for every pixel. i.e., FMT supposes that the translation $(x_\delta, y_\delta)$ and zoom $z_\delta$ is the same for all pixels $p$. Thus for FMT all $(x_\delta, y_\delta)$ fall in a singe cell, forming a peak.

In this paper, we propose eFMT that relaxes the equidistance constraint by solving (10) with different depths $z_\delta$ to estimate camera poses.

## 4. Methods

In this section, we first solve (10) with the translation-only case and zoom-only case, respectively. Then we present how to handle the general case with 4DoF motion. Since the absolute magnitude of the monocular camera's poses cannot be found, the re-scaling for translation and zoom is also discussed to estimate the up-to-scale transformation.

Without loss of generality, we use frame indices 1, 2 and 3 for any three consecutive frames in this paper.

### 4.1. Translation-Only Case

FMT decouples the translation estimation from rotation and zoom calculation. Thus we only consider that the camera moves in the $x - y$ plane in the translation-only case. Then Equation (10) is simplified to

$$I_2(x, y) = I_1(x + x_\delta, y + y_\delta) \quad . \tag{14}$$

As indicated by Equations (12) and (13), translation $(x_\delta, y_\delta)$ is not a single energy peak in the PSD as in Equation (9), due to the multi-depth environment. Figure 2 shows a translation PSD in the multi-depth environment. It can be seen that there are multiple peaks in the PSD and the $x - y$ view shows that these high peaks lie on one line. The collinear property is derived from the definition of $x_\delta$ and $y_\delta$. In the translation-only case, Equations (12) and (13) are reduced to

$$x_\delta = \frac{f_x \Delta x}{\delta}, y_\delta = \frac{f_y \Delta y}{\delta} \ .$$

It can be found that the direction of each translation $(x_\delta, y_\delta)$ is the same, i.e.,:

$$\left( \frac{f_x \Delta x}{\sqrt{(f_x \Delta x)^2 + (f_y \Delta y)^2}}, \frac{f_y \Delta y}{\sqrt{(f_x \Delta x)^2 + (f_y \Delta y)^2}} \right),$$

which is independent on the pixel depth $\delta$. Also, the translation $(x_\delta, y_\delta)$ lies in the line:

$$f_y \Delta y \cdot x - f_x \Delta x \cdot y = 0 \ .$$

Thus, the peaks with high values lie in a line across the center of the PSD. Additionally, pixels cannot move in the opposite direction. So the peaks lie in a line that starts from the center. The extreme case is a slanted plane in the camera's view. Then there are not distinguishable peaks, but a continuous line segment in the PSD. To keep it general and don't rely on peak detection, this paper proposes the following way to estimate the translation.

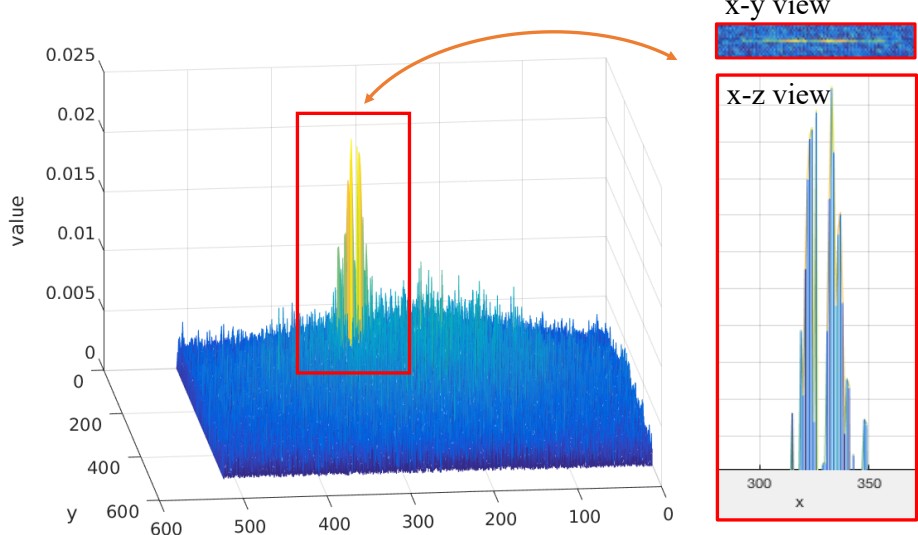

**Figure 2.** An example of a translation phase shift diagram in a multi-depth (more than two depths) environment.

Independent of their depth, with a given camera translation, all pixels will move with collinear translation vectors - the magnitude of this translation depends on their depth and

the magnitude of the camera translation. Thus we can treat the translation estimation in a novel way different from finding only the highest peak. Concretely, starting from the center of the PSD, which represents the no-translation case, we perform a polar search for the sector $r_{max}$ that sums up the most energy. This sector now represents the direction of the translation vector, abbreviated as $t$. We have no concrete estimate for the magnitude of the motion, which would be anyways up to the unknown scale factor, therefore the estimated translation vector $t$ is a unit vector, which is called ***unit translation vector*** in this paper.

As introduced in Section 1, one weakness of FMT is that it does not consider the scale consistency for visual odometry, where the estimated translation between images $I_1$ and $I_2$ has to be re-scaled to be in the same unit as the one between $I_2$ and $I_3$. To overcome this drawback, eFMT calculates the re-scaling factor on the $r_{max}$ sector. For that, we sample a translation energy vector $\mathbb{V}_t$ from the $r_{max}$ sector of the PSD. With a given camera translation, regions with different depths correspond to different indices in the translation energy vector. The more pixels correspond to a region, the higher the energy. Assume the translation energy vector between $I_1$ and $I_2$ is $_1^2\mathbb{V}_t$ and that between $I_2$ and $I_3$ is $_2^3\mathbb{V}_t$. The second image $I_2$ is shared between both translations, thus the depths of the regions are the same for both translations. Any difference between the translation energy vectors $_1^2\mathbb{V}_t$ and $_2^3\mathbb{V}_t$ must thus come from different magnitudes of translation, independently from the direction of that translation. In fact, the vectors are simply scaled by the ratio of the translation magnitudes, which then also maintains the correspondence of the regions and their size/ energy values in the vectors. Thus, the re-scaling factor $_{3\to2}^{2\to1}s_t$ can be calculated via pattern matching on $_1^2\mathbb{V}_t$ and $_2^3\mathbb{V}_t$ by

$$_{3\to2}^{2\to1}s_t = \arg\min_s ||_1^2\mathbb{V}_t - f(_2^3\mathbb{V}_t, s)||_2^2 \,, \tag{15}$$

where $f(\cdot)$ uses $s$ to scale the vector $_2^3\mathbb{V}_t$ in length and value. Details are presented in Section 5.

Differences in the regions from changing occlusions and field of views add noise to the PSD but can be ignored in most cases, analogous to the image overlap requirement in the classical FMT [39].

*4.2. Zoom-Only Case*

As implied in Equation (4), rotation and zoom share the same PSD (see Figure 3). Also, the rotation is depth-independent and the same for all pixels, as shown in Equation (10). Thus, eFMT calculates rotation in the same way as FMT does. In this section, we just consider the zoom-only case, i.e., the camera moves perpendicular to the imaging plane. In this case, the Equation (10) is simplified to

$$I_2(x, y) = I_1(z_\delta x, z_\delta y)) \,. \tag{16}$$

Meanwhile, Equation (4) becomes

$$M_2(\xi, \theta) = M_1(\xi - d_\delta, \theta) \,, \tag{17}$$

where $d_\delta = \log z_\delta$. Therefore, the multiple peaks of zoom lie in one column in the rotation and zoom PSD, because all the zoom peaks correspond to one rotation, i.e., the same column index. Note that these zoom peaks are sometime continuous in real applications due to the continuous depth change, then these zoom peaks become high values in the PSD. For that, we no longer search for multiple peaks. Instead, a set of multi-zoom values $\mathbb{Z} = \{z_\delta\}$ is uniformly sampled between the maximum zoom $z_{max}$ and minimum zoom $z_{min}$ estimated from the column $\mathbb{C}_z^*$ with maximum sum energy. $\mathbb{C}_z^*$ can be found by

$$\mathbb{C}_z^* = \arg\max_{\mathbb{C}_z}\{\mathbb{C}_z \in q_z\} \,, \tag{18}$$

where $q_z$ is the rotation and zoom PSD. Then we find the highest peak value $E_{max}$ of the column $\mathbb{C}_z^*$. Only the values whose energy is larger than half $E_{max}$ are called high values in $\mathbb{C}_z^*$. The maximum zoom $z_{max}$ and minimum zoom $z_{min}$ are searched from these high values by calculating zooms from the index of the high values. In addition, as derived in Section 3, the zoom $z_\delta$ is described by Equation (11), which is inversely proportional to the depth $\delta$. Thus, the minimum and maximum zooms, estimated from the PSD, indicate the maximum and minimum pixel depths, respectively. Since the energy in the translation PSD also relates to the pixel depths, we can build correspondences between zoom energy and translation energy, which will be discussed in the next section.

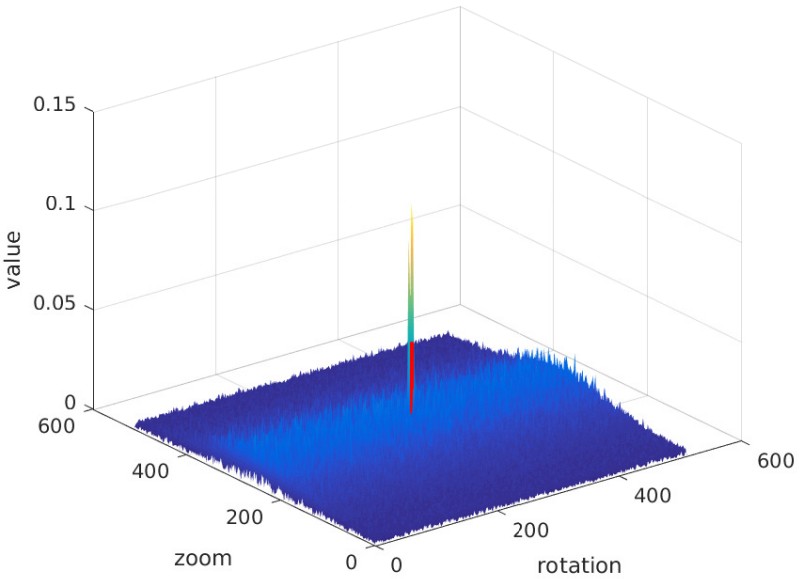

**Figure 3.** An example of rotation and zoom phase shift diagram.

Additionally, re-scaling for zoom is also essential in the zoom-only case for visual odometry. For that, a zoom energy vector $\mathbb{V}_z$ is extracted from $\mathbb{C}_z^*$. $\mathbb{V}_z$ is the half of $\mathbb{C}_z^*$ with higher energy. This is based on the prior knowledge that all regions should consistently either zoom in or out. Suppose the zoom energy vector between $I_1$ and $I_2$ is $_1^2\mathbb{V}_z$ and that between $I_2$ and $I_3$ is $_2^3\mathbb{V}_z$. The re-scaling factor $_{3\to2}^{2\to1}s_z$ between $_1^2\mathbb{V}_z$ and $_2^3\mathbb{V}_z$ is found by

$$_{3\to2}^{2\to1}s_z = \arg\min_{s} ||_1^2\mathbb{V}_z - g(_2^3\mathbb{V}_z, s)||_2^2 , \tag{19}$$

where $g(\cdot)$ is the function of shifting the vector $_2^3\mathbb{V}_z$. It is a variant of the pattern matching used in translation re-scaling. The only difference is that, while the translation energy vectors above are matched via scaling, the zoom energy vectors must be matched via shifting. Both algorithms will be shown in Section 5.

### 4.3. General 4DoF Motion

When the 4DoF motion of the camera happens, the transformation between two poses is estimated following the scheme of the FMT. Our eFMT pipeline is shown in Figure 4. Since monocular visual odometry algorithms are up-to-scale [1], we use three frames to calculate the up-to-scale transformation.

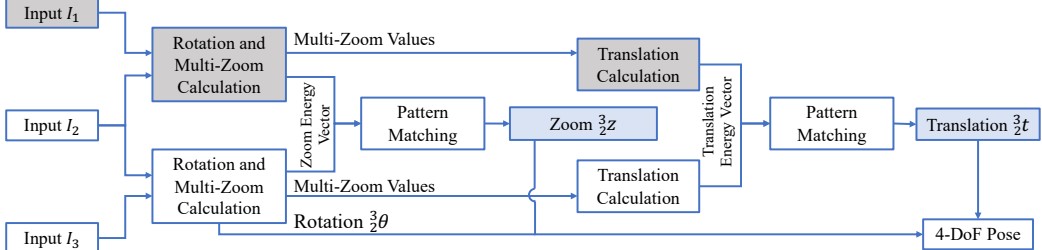

**Figure 4.** Pipeline of eFMT. Please note that the output is the 4DoF transformation between frame $I_2$ and $I_3$, but using 3 frames to estimate the re-scaling factor. The gray boxes indicate that the computation results of the previous iteration are reused.

Similar to the FMT pipeline [18], we firstly calculate the rotation and zoom between two frames. Instead of searching for the highest peak value on the rotation and zoom PSD, we exploit all the information of half a column of the PSD in eFMT, yielding multi-zoom values $\mathbb{Z} = \{z_\delta\}$ and the zoom energy vector $\mathbb{V}_z$, as introduced in Section 4.2. In addition, the multi-zoom values $\mathbb{Z} = \{z_\delta\}$ are uniformly sampled between the minimum and maximum high zoom values of the PSD, which takes the energy instead of peaks into consideration. Thus it is robust to the continuous energy in the PSD. Afterwards, we obtain translation PSDs for the rotation $\theta_0$ and each zoom value $z_\delta$, by first re-rotating and re-zooming the second image:

$$I_2' = I_2(z_\delta x \cos\theta_0 - z_\delta y \sin\theta_0, z_\delta x \sin\theta_0 + z_\delta y \cos\theta_0) \, ,$$

and then performing phase correlation on image $I_1$ and $I_2'$ with Equation (7). With the method introduced in Section 4.1, the translation energy vector $\mathbb{V}_{t,z_\delta}$ is extracted from the translation PSD. Then these multiple translation energy vectors are combined according to the weight of the zoom energy:

$$\substack{2\\1}\mathbb{V}_t = \sum_{z_\delta \in \mathbb{Z}} \frac{\mathbb{V}_z[z_\delta]}{U} * \substack{2\\1}\mathbb{V}_{t,z_\delta} \, , \tag{20}$$

where $\mathbb{V}_z[\cdot]$ is the function to find the energy corresponding to the zoom $z_\delta$ and $U = \sum_{z_\delta \in \mathbb{Z}} \mathbb{V}_z[z_\delta]$. Since the higher the zoom value is, the more pixels correspond to the zoom, the corresponding translation energy vector should get the higher weight accordingly. Thus the Equation (20) holds.

### 4.4. Tidbit on General 4DoF Motion

Classical FMT decouples rotation and zoom from the translation. For eFMT this is not as simple: as the camera moves along the z-axis (perpendicular to the image plane), objects of different depth are zoomed (scaled) by different amounts. In a combined zoom and translation case, the apparent motion of a pixel depends on its depth, the zoom and translation. However, for the pattern matching of the translation energy vectors (Equation (15)) to be based just on a simple scaling, the energy in the pixel motions has to be based just on the pixel depth and translation speed, so they must be independent of the zoom. As described above, eFMT will calculate translation energy vectors $\mathbb{V}_{t,z_\delta}$ for different zoom values. This means that in multi-depth images there will be parts of the image that are zoomed with the incorrect zoom value but are then used as input in Equation (7) and ultimately combined into the translation energy vector from Equation (20).

One could assume that those incorrectly zoomed image parts lead to wrong pixel translation estimations, thus leading to a compromised translation energy vector. However, this is not the case: The phase correlation (Equation (7)) is sensitive to the zoom! It will only notice signals that are in the same zoom (scale)—other parts will just be noise. This is because with a wrong zoom Equation (14) does not hold. Figure 5 shows how a wrong zoom will influence the translation PSD. It can be found that wrong zoom decreases the

energy of the correct translation and distributes the energy over the PSD. Also, a slight difference does not change the translation PSD too much, whereas a big difference will result in a PSD with mostly uniformly distributed noise. To give a better explanation, we also demonstrate the signal-to-noise ratio (SNR) of the translation PSD with different fixed zoom values in Figure 6. The SNR value is calculated by the ratio between the mean of the high values from the translation energy vector and the mean of the remaining values in the PSD. Figures 5 and 6 show that a deviation of zoom of only 0.08 will lead to a SNR below 2.6, which is very noisy already. Thus when eFMT is iterating through the different multi-zoom values in $\mathbb{Z}$, the translation energy vectors $\mathbb{V}_{t,z\delta}$ will just notice the pixels that are correctly zoomed, because the wrongly zoomed values will be very small compared to the actual high values with the right zoom. Thus the combined translation energy vector $\mathbb{V}_t$ is independent of the zoom. Therefore, also the pattern matching of the translation energy vectors for re-scaling is zoom independent.

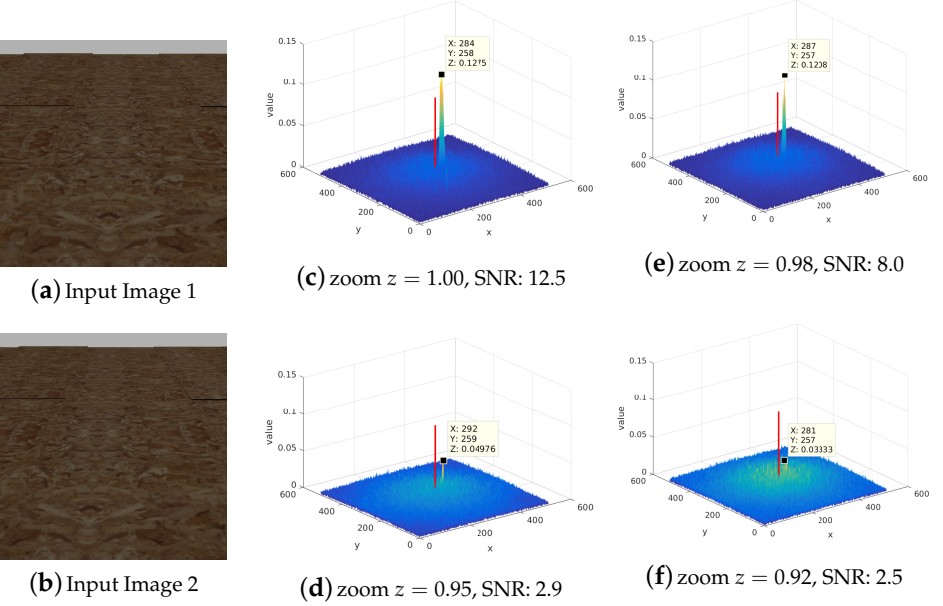

**Figure 5.** Translation PSDs with different zoom values. The groundtruth zoom between the two input images is 1. To test the influence of zoom on the translation phase shift diagram, we set zoom value from 1.00 to 0.92 manually to re-zoom the second image and then perform phase correlation.

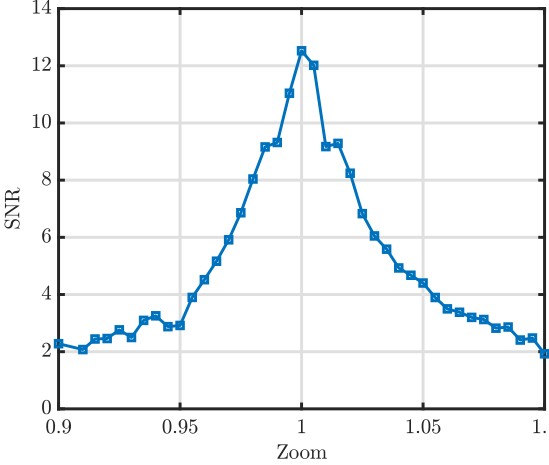

**Figure 6.** Signal-to-Noise ratio with different zoom. (Correct zoom is 1).

### 4.5. Practical Consideration—Visual Odometry

We demonstrate the advantage of eFMT over FMT on camera pose estimation, i.e., visual odomtery. The main considerations in visual odometry are how to put translation and zoom in the same metric, i.e., translation and zoom consistency.

For that, we analyze the relationship between image transformation and camera motion again. As shown in Figure 7, assume the objects with size $l_i$ and depth $\delta_i$ are in the FOV of the camera $C$ in Pose 1. The camera moves to Pose 2 with the motion $(\Delta x, \Delta y)$ in the $x - y$ plane and $\Delta\delta$ along the $z$ direction.

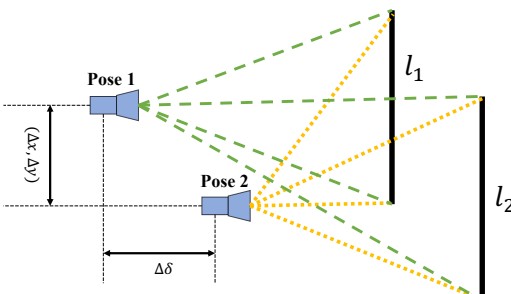

**Figure 7.** Objects of different depths in the FoV of the camera. When the camera moves from Pose 1 to Pose 2, the pixel motions of $l_1$ and $l_2$ are inversely proportional to the depth.

According to the basic properties of pinhole cameras, the zoom between the two frames captured in Pose 1 and 2 is $z_{\delta_i} = \frac{\delta_i}{\delta_i + \Delta\delta}$. Similarly, we can derive the translation between two frames of different depths $\delta_j$. We are using $j$ here, because in the algorithm, zoom and translation are calculated independently. The pixel translation between Pose 1 and 2 are $(\frac{f\Delta u_j}{\delta_j}, \frac{f\Delta v_j}{\delta_j})$, where $f$ is the focal length of the camera. Then the ratio between the translation perpendicular to the imaging plane and that in the $x - y$ plane can calculated by

$$\frac{(\frac{1}{z_{\delta_i}} - 1)f}{\|(\Delta u_j, \Delta v_j)\|} , \tag{21}$$

if and only if $i = j$, meaning that the same object distance $\delta = \delta_i = \delta_j$ is used.

We can use pattern matching between the zoom energy vector and the translation energy vector to find the corresponding $i$ and $j$. For simplicity, in this paper we use maximum energy finding to determine the zoom $z_{peak}$ with the highest peak in the zoom energy vector $\mathbb{V}_z$ (this corresponds to $l_i$ with depth $\delta_i$). In the translation energy vector for $z_{peak}$ we then find the peak translation vector $(\Delta u', \Delta v')$ ($l_j$, which actually is $l_i$). This holds for all pixels with the same depth without the limitation of lying in one continuous plane.

Then we can get the 3D translation $t$ between the camera poses:

$$t = \begin{bmatrix} \Delta x \\ \Delta y \\ \Delta\delta \end{bmatrix} = \begin{bmatrix} \Delta u \\ \Delta v \\ \frac{(\frac{1}{z_{peak}} - 1)f}{\|(\Delta u', \Delta v')\|} \end{bmatrix} , \tag{22}$$

where $(\Delta u, \Delta v)$ is the unit translation vector.

### 4.6. Summary of Key Ideas

The key ideas of eFMT are outlined as follows:

- Observation that multiple depths will lead to multiple strong energies in the PSDs for zoom and translation, and that these signals are collinear.
- Instead of finding one maximum peak, as the classical FMT is doing, we represent the translation in a one-dimensional translation energy vector that encodes the number of pixels with certain amounts of motion, which correspond to certain depths. We treat

the orientation and the magnitude independently. The orientation from the center of the PSD, from which the translation energy vector was sampled, is the direction of the motion, represented as a unit translation vector. The zoom is represented analogous. Thus, eFMT keeps the accuracy and robustness of FMT w.r.t features and direct methods, and improves the scale consistency of FMT.

- We put the zoom and translation in the same reference frame by finding the correspondence between zoom and translation based on pattern matching.
- Finally, we assign a magnitude to the second of the two found unit translation vectors of three consecutive frames by estimating a re-scaling factor between the translation energy vectors via pattern matching. The re-scaling for zoom is estimated analogously.

## 5. Implementation

This section introduces the implementation of a visual odometry framework based on eFMT. We first present this framework and then discuss in detail how to implement re-scaling for translation and zoom.

Algorithm 1 demonstrates the implementation of the eFMT-based visual odometry. FMT is directly applied for the first two frames to estimate rotation $\theta_0$, zoom $z$ and unit translation vector $t$. Additionally, zoom and translation energy vectors $\mathbb{V}_z$ and $\mathbb{V}_t$, used for pattern matching in the next iteration, are generated from the corresponding PSDs, respectively. For the following frames, eFMT is performed to calculate re-scaled zoom and translation, so that the 4DoF motion between frames can be estimated. Moreover, the trajectory of the camera is generated via the chain rule.

---

**Algorithm 1** eFMT-based Visual Odometry

---

1: *Input:* $\mathbb{I} = \{I_i | i \in \mathbb{N} \wedge 0 \leq i < \# \text{ frames}\}$
2: **for** $i$ in $[1..len(\mathbb{I})]$ **do**
3:　　**if** $i = 1$ **then**　　　　　　　　　　　　　　　　　　　　▷ Similar to FMT
4:　　　　Estimate rotation $_0^1\theta_0$, zoom $_0^1z$ and translation $_0^1t$
5:　　　　Generate $_0^1\mathbb{V}_z$ from the rotation and zoom PSD
6:　　　　Generate $_0^1\mathbb{V}_t$ from the translation PSD
7:　　**else**　　　　　　　　　　　　　　　　　▷ Multi-zoom and Multi-translation
8:　　　　Calculate the rotation and zoom PSD between
　　　　　　$I_{i-1}$ and $I_i$
9:　　　　Estimate the rotation $_{i-1}^{i}\theta_0$ and zoom values
　　　　　　vector $_{i-1}^{i}\mathbb{Z}$ from the PSD
10:　　　　Generate $_{i-1}^{i}\mathbb{V}_z$ from the PSD
11:　　　　**for** $j$ in $[0..len(_{i-1}^{i}\mathbb{Z})]$ **do**
12:　　　　　　Get translation energy vector $_{i-1}^{i}\mathbb{V}_{t,j}$ and
　　　　　　　　unit translation vector $_{i-1}^{i}t_j$
13:　　　　**end for**
14:　　　　Combine translation energy vector to $_{i-1}^{i}\mathbb{V}_t$
　　　　　　via (20)
15:　　　　Estimate 3D translation introduced in Section 4.5
16:　　　　Estimate re-scaling factor between $_{i-2}^{i-1}\mathbb{V}_z$ and
　　　　　　$_{i-1}^{i}\mathbb{V}_z$ via pattern matching
17:　　　　Estimate re-scaling factor between $_{i-2}^{i-1}\mathbb{V}_t$ and
　　　　　　$_{i-1}^{i}\mathbb{V}_t$ via pattern matching
18:　　　　Update zoom and translation
19:　　　　Perform chain rule on the 4 DoF transformation
20:　　**end if**
21: **end for**
22: *Output:* camera poses corresponding to $\mathbb{I}$

---

As described above, for translation calculation, we find the sector with maximum sum energy $r_{max}$ instead of the highest peak. Concretely, the PSD is divided into $n$ sectors from the center $b$. Then we sum up the energy of the cells in each sector within a certain opening angle $o$, e.g., $2°$, to find the $r_{max}$. Afterwards, the direction from the center $b$ to the highest value of the sector $r_{max}$ is considered to be the translation direction, i.e., unit translation vector ${}_1^2 t_i$. Furthermore, we represent the values of the maximum sector ${}_1^2 r_{max}$ as the 1D energy vector ${}_1^2 \mathbb{V}_{t,z_\delta}$. We sample the energy in the maximum sector $r_{max}$ at uniform distances to fill ${}_1^2 \mathbb{V}_{t,z_\delta}$. Then the translation energy vectors are combined to ${}_1^2 \mathbb{V}_t$ with Equation (20).

Moreover, the pattern matching algorithms used in the re-scaling for translation and zoom (Equations (15) and (19)) are shown in Algorithms 2 and 3, respectively. Equations (15) and (19) are in the form of least-squares problems, which are often solved by gradient decent methods. However, since the explicit expression of function $f(\cdot)$ in Equation (15) is pointwise on the variable $s$, it is difficult to construct the Jacobian when using the gradient decent methods to solve Equation (15). Additionally, the gradient decent method is prone to local minima, especially without a good initial guess. In fact, our method does not provide any initial guess. Solving Equation (19) is analogous. Therefore, the pattern matching Algorithms 2 and 3 are exploited to find re-scaling factors in this work. There are several methods to handle pattern matching, for example phase correlation, search algorithms and dynamic programming. Considering the robustness on outliers of the PSD signals, we use a search method in this paper.

---

**Algorithm 2** Re-scaling for Translation

1: *Input:* ${}_1^2 \mathbb{V}_t$ and ${}_2^3 \mathbb{V}_t$
2: Initialize distance $d$ with infinity
3: **for** $s = 0.1 : 0.002 : 10.0$ **do**
4:     Scale ${}_2^3 \mathbb{V}_t$ to ${}_2^3 \mathbb{V}_t'$ with $s$
5:     Calculate Euclidean distance $d_s$ between ${}_1^2 \mathbb{V}_t$ and ${}_2^3 \mathbb{V}_t'$
6:     **if** $d_s < d$ **then**
7:         $d \leftarrow d_s$
8:         ${}_{3 \to 2}^{2 \to 1} s_t \leftarrow s$
9:     **end if**
10: **end for**
11: *Output:* rescaling factor ${}_{3 \to 2}^{2 \to 1} s_t$

---

**Algorithm 3** Re-scaling for Zoom

1: *Input:* ${}_1^2 \mathbb{V}_z$ and ${}_2^3 \mathbb{V}_z$
2: Initialize distance $d$ with infinity
3: **for** $\Delta = -r : 1 : r$ **do**            $\triangleright$ $r$ is the length of ${}_1^2 \mathbb{V}_z$
4:     Shift ${}_2^3 \mathbb{V}_z$ to ${}_2^3 \mathbb{V}_z'$ with $\Delta$
5:     Calculate Euclidean distance $d_s$ between ${}_1^2 \mathbb{V}_z$ and ${}_2^3 \mathbb{V}_z'$
6:     **if** $d_s < d$ **then**
7:         $d \leftarrow d_s$
8:         ${}_{3 \to 2}^{2 \to 1} s_z \leftarrow$ shift_to_scale$\{\Delta\}$
9:     **end if**
10: **end for**
11: *Output:* rescaling factor ${}_{3 \to 2}^{2 \to 1} s_z$

---

## 6. Results

In this section, we evaluate the proposed eFMT algorithm in both simulated and real-world multi-depth environments. Note again that there are multiple variants of FMT,

we use the improved one from [19,20] for better robustness and accuracy. Since all the FMT implementations only search for one peak in the PSDs, they will meet difficulties in multi-depth environments, no matter which implementation is used.

We first present basic experiments about the zoom and translation re-scaling in the simulation test. The scenario only includes two planes with different depths to show the basic effectiveness of eFMT. Then eFMT is compared with FMT and the state-of-the-art VO methods, ORB-SLAM3 [4], SVO [5] and DSO [7], in the real-world environments. The three state-of-the-art VO methods that do not rely on FMT are the most popular and representative monocular ones, as pointed out in [38]. The tests in the real-world environments include two parts: one toy example with two wooden boards and a large-scale UAV dataset (https://robotics.shanghaitech.edu.cn/static/datasets/eFMT/ShanghaiTech_Campus.zip, accessed on 5 March 2021). The toy example is similar to the simulation environment. Since the features are very similar on the wooden board, the scenario is more difficult than general indoor environments, even though there are only two planes. To evaluate the eFMT algorithm in a more general case and provide a potential use-case of eFMT, we proceed the second test with a down-looking camera mounted on a UAV. The scenario includes many different elements, such as building roofs, grass and rivers. Since there are many different depths in the view, especially that the building will be a slanted plane due to the perspective projection, it is thus challenging for FMT. In addition, the feature-deprived road surface and grass would be a big challenge for classic VO methods. We will show that eFMT can handle both difficulties.

All experiments are conducted with an Intel Core i7-4790 CPU@3.6 GHz and 16 GB Memory without GPU. The algorithm is implemented in C++ using a single thread.

### 6.1. Experiments on the Simulated Datasets

In this test, images are collected in the Gazebo simulation for accurate ground truth. As shown in Figure 8, the camera is equipped on the end-effector of a robot arm such that we can control the robot arm to move the camera.

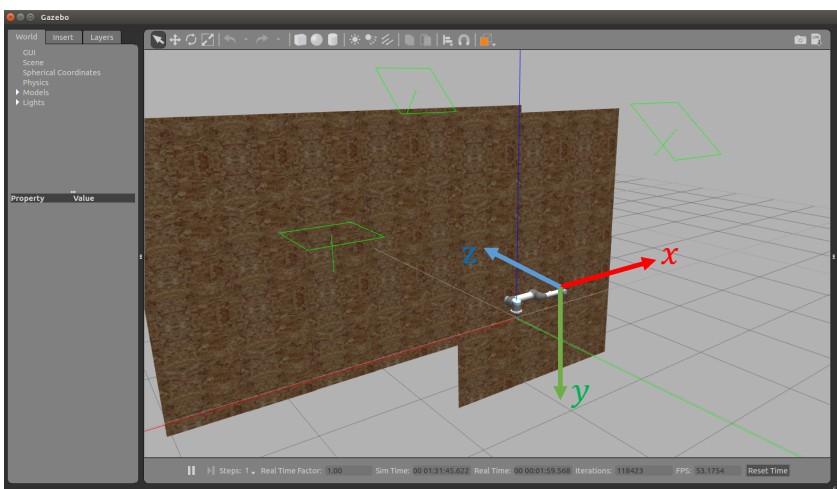

**Figure 8.** Simulated environment (y points down). The camera is equipped on the end-effector of a robot arm such that we can control the robot arm to move the camera.

### 6.1.1. Zoom Re-Scaling

In this case, we move the robot arm along the *z*-axis to generate three simulated images with two planes in different depths. As shown in Figure 9b–d, they are zoomed in from left to right. In each image, the left half is further whereas the right half is closer. Then the rotation and zoom PSDs are shown in the second row of Figure 9. It can be seen that each diagram has two peaks, which indicates two different depths in the view. Moreover, the higher peak is not always in the left, which implies the majority depth in the view changes, which destroys the scale consistency of the FMT. Traditional FMT only uses the

highest peak. Instead, the proposed eFMT takes the zoom energy vector into consideration and puts all zoom values into the same scale through re-scaling—up to one unknown scale factor.

Here we show that the eFMT outperforms FMT by using the three images as a small loop closure. The zoom ${}_0^2z$ between image 0 and 2 should equal to the product of the zoom ${}_0^1z$ between image 0 and 1 and ${}_1^2z$ between image 1 and 2. The result in Table 1 shows that eFMT estimates the zoom correctly, so that the zoom loop holds, i.e., $||{}_0^1z * {}_1^2z/{}_0^2z|| \approx 1$. However, FMT only tracks the highest peak. The plane that the highest peak in Figure 9g corresponds to is different from that in Figure 9e,f, so ${}_0^2z$ and ${}_1^2z$ are calculated based on different planes with different depths. Thus $||{}_0^1z * {}_1^2z/{}_0^2z||$ is further away from 1.

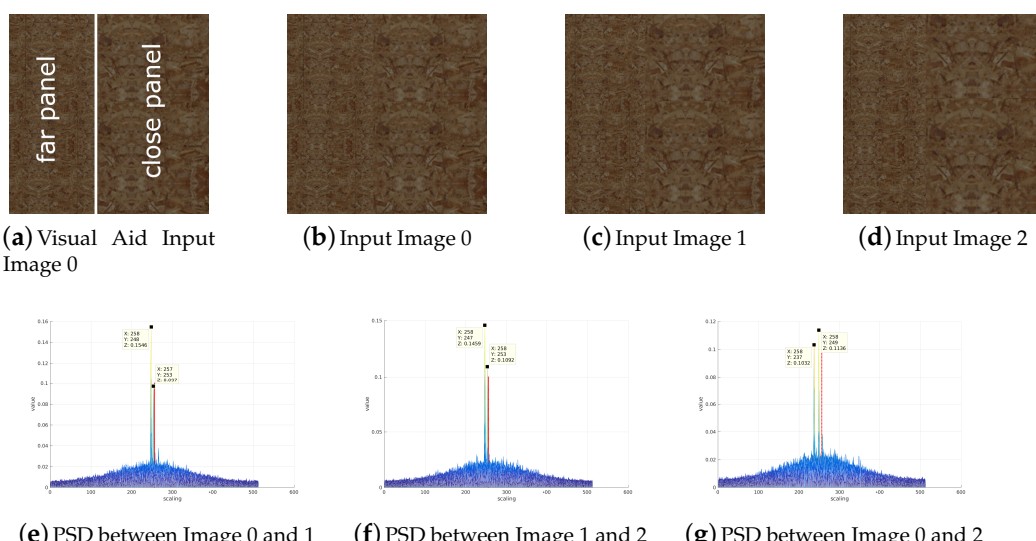

(**a**) Visual Aid Input Image 0　(**b**) Input Image 0　(**c**) Input Image 1　(**d**) Input Image 2

(**e**) PSD between Image 0 and 1　(**f**) PSD between Image 1 and 2　(**g**) PSD between Image 0 and 2

**Figure 9.** Three rotation and zoom phase shift diagrams (PSD) with multiple zooms. In the first row, the left parts of the input images are further than the right parts w.r.t the camera. Thus there are two peaks in the rotation and zoom PSD.

**Table 1.** Loop Closure for Zoom Estimation.

|  | ${}_0^1z$ | ${}_1^2z$ | ${}_0^2z$ | $||{}_0^1z * {}_1^2z/{}_0^2z||$ |
|---|---|---|---|---|
| eFMT | 0.889 | 0.889 | 0.768 | 1.029 |
| FMT [19,20] | 0.889 | 0.881 | 0.902 | 0.868 |

### 6.1.2. Visual Odometry in Simulated Scenario

In this case, the simulated robot arm moves in the $x - z$ plane to generate images with combined translation and zoom. Here, we compare the visual odometry based on eFMT and FMT on this dataset. Figure 10 shows that eFMT tracks the correct re-scaling factor to the end while the FMT fails at about $z = -0.5$ m, which indicates that eFMT also works better than FMT with zoom and translation. This benefits from the re-scaling based on pattern matching, as introduced in Section 4.

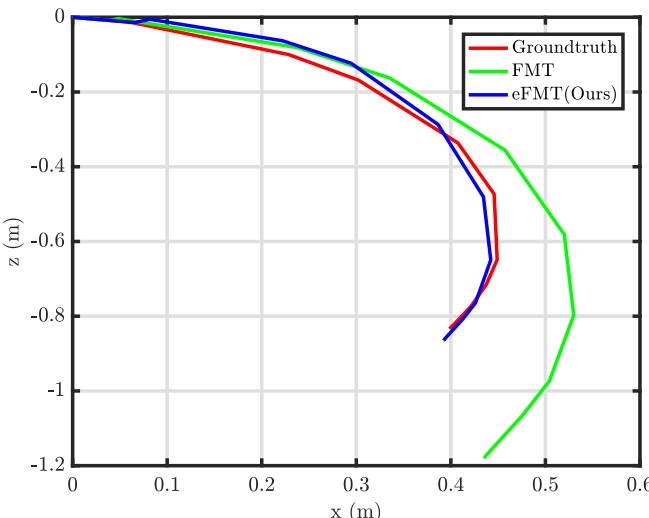

**Figure 10.** Visual odometry comparison in a simulated scenario. The data is collected with the setting in Figure 8.

### 6.2. Experiments on Real Datasets

After the preliminary tests in the simulated environment, we evaluate the performance of eFMT by comparing with FMT and other state-of-the-art VO methods in real-world scenarios. The first example is similar to the simulation setting with two wooden boards in the camera's view, as shown in Figure 11a. The ground truth is provided by a tracking system. In the second example, we collect a dataset with an unmanned aerial vehicle flying over our campus. More details are introduced in the following.

#### 6.2.1. A Toy Example

In this case, we evaluate the visual odometry with only translation along the $x-$axis (see in Figure 11a) with two different depths. Similar to the simulation, the wooden board with smaller depth first goes into the camera's view, then both boards are in the view, finally only the wooden board with larger depth is observed.

Figure 11b compares the localization results with different methods, including FMT (green triangle), eFMT (blue star), SVO (blue triangle) and ORB-SLAM3 (brown star). The results of DSO are omitted here because it fails tracking in this scenario. To compensate the unknown scale factor, the estimated results are aligned to the ground-truth (via a tracking system) by manual re-scaling. Since the camera only moves in the $x$ direction, we only show the positions in $x$ axis versus frames. The absolute error (Table 2) will include errors in both $x$ and $y$ direction.

**Table 2.** Absolute trajectory error comparison.

| Methods | Mean (mm) | Max (mm) | Median (mm) |
|:---:|:---:|:---:|:---:|
| FMT [19,20] | 17.1 | 54.7 | 10.1 |
| eFMT | **2.1** | **6.0** | **1.8** |
| SVO [5] | 17.0 | 38.0 | 17.5 |
| ORB-SLAM3 [4] | 13.0 | 21.6 | 13.6 |
| DSO [7] | / | / | / |

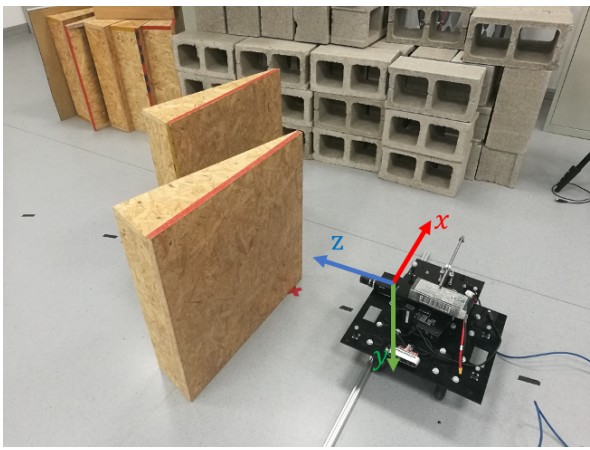

(**a**) Real environment (y points down)

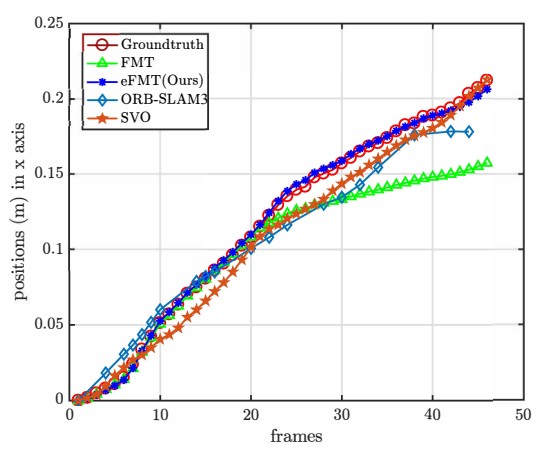

(**b**) Visual Odometry Comparison

**Figure 11.** A visual odometry example in a real-world environment. DSO fails in this test, thus it is ignored in this figure.

We can see that FMT begins to suffer from scale drift approximately from the 20th frame, where FMT changes the tracked panel, because the new panel now is bigger in the view and thus has a higher peak. That new panel is further away, thus the pixels move slower, thus FMT underestimates the motion compared to previous frames. In contrast, the proposed eFMT maintains the correct scale till the last frame, because our pattern matching re-scales all unit translation vectors correctly. Compared with SVO and ORB-SLAM3, eFMT tracks each frame more accurately. The absolute trajectory errors in Table 2, including mean, max and median of errors, also shows that eFMT achieves the smallest error, followed by SVO and ORB-SLAM3. Concretely, the mean error of eFMT is approximately 1/6 of that of ORB-SLAM3, and about 1/8 against FMT or SVO. This test shows that eFMT outperforms the popular visual odometry algorithms in this challenging environment, thanks to the robustness of the spectral-based registration.

### 6.2.2. The UAV Dataset

In addition to the above toy examples, we compare the proposed eFMT with FMT, ORB-SLAM3, SVO and DSO on a bigger UAV dataset. Note that even though there are several public UAV datasets [40–44], we could not use them in this paper because we require datasets without roll/ pitch due to the properties of our algorithm. Ref. [41] provides such a dataset, NPU dataset, with no roll/pitch, but the flying height of the UAV is too high, so that the scenario can be considered single-depth. We tested on one sequence of the NPU dataset, which the UAV collected over farmland. Since the features in

this scenario are ambiguous, ORB-SLAM3, SVO and DSO failed to estimate the camera trajectory on this dataset. Both FMT and eFMT succeed to track the trajectory with some accumulated error, and their performance are similar due to single depth. The algorithm presented in [41] works well on this data, but it is using GPS in the algorithm, it assumes a planar environment and is a fully fledged SLAM system, matching against map points, while eFMT is just registering two consecutive image frames.

To show the performance of eFMT in a multi-depth setting, we collect a dataset (https://robotics.shanghaitech.edu.cn/static/datasets/eFMT/ShanghaiTech_Campus.zip, accessed on 5 March 2021), which is released together with this paper. Our dataset is collected by a down-looking camera equipped on a DJI Matrice-300 RTK. The flying speed is set to 2 m/s and the image capture frequency is 0.5 Hz. The path of the drone over our campus is shown in Figure 1. The DJI aerial vehicle collected 350 frames on a trajectory of about 1400 m. The height above ground is about 80 m, which is approximate 20 m higher than the highest building. As we mentioned in the beginning of the experiment, this dataset contains the all kinds of different elements. These include roofs, road surfaces, a river and grass, where some of them are challenging for the classic VO methods that are not based on FMT. Furthermore, the multiple depths increase the difficulty for FMT. In this case, we will show that the eFMT not only keeps the robustness of FMT but also overcomes its single-depth limitation.

The overall trajectories of different approaches are shown in Figure 12. The trajectories are aligned with a scale a rotation calculated from the poses of the 0th frame and the 80th frame. We refer readers to the attached Supplementary Video for the frame-by-frame results. Since SVO and DSO fail to estimate the camera poses, the trajectories of them are not included in this figure. Also, it can be found that the ORB-SLAM3 fails to track several times, as indicated by the red stars. After each failure, the trajectory of ORB-SLAM3 is realigned. Both FMT and eFMT succeed to estimate the camera poses till the end of the dataset, though the translation has some drift. To evaluate the performance of FMT, eFMT and ORB-SLAM3, we compare these methods only up to the frame that ORB-SLAM3 fails. The performances of different approaches are shown in Figure 13. From the right local enlarged figure, we can find that the estimated speeds of eFMT and ORB-SLAM3 are almost constant, as indicated by the equal distances between the frames. This is consistent with the centimeter-grade RTK GPS ground truth. However, the estimated speed of FMT changes according to the view. For instance, the speed is faster from frame 125 to 132 than that from frame 132 to 138, because the dominate plane is ground in the former case whereas the dominant plane changes to the roof in the latter case. In addition, Figure 14 displays the absolute translation error versus distances with the evaluation tool from [45]. If only comparing the performance when all three approaches are tracking successfully, the performance of eFMT is on a par with ORB-SLAM3 and both of them are better than FMT, because FMT suffers from different depths.

Please note that there are continuous line segments in the translation PSD when there are slanted planes in the view. As shown in Figure 15, the buildings in Image 1 and 2 become inclined due to the perspective projection, which yields the line segments (left to the red center) in the translation PSD below. In the UAV dataset, such inclined planes are common, thus pattern matching is necessary for re-scaling. Moreover, the estimated trajectory shown in Figure 12 shows that eFMT can handle such slanted planes issues.

Thus, this experiment shows that eFMT has two advantages: (1) it successfully extends FMT to multi-depth environments, that is, no matter the multiple depths are continuous (e.g., slanted plane) or discrete (e.g., roofs and ground), eFMT can track the camera motion; (2) it keeps the robustness of FMT that it can still track the camera motion in the feature-deprived scenarios, such as building roofs, whereas the classic VO methods may fail tracking. The experiment mentioned in the beginning of Section 6.2.2 also supports the second point.

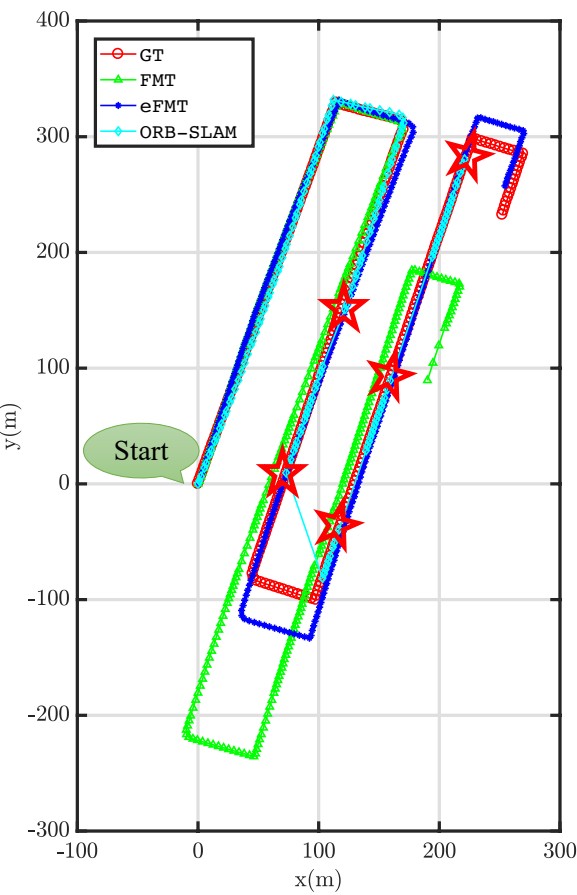

**Figure 12.** Overall trajectories of different methods on the UAV dataset. ORB-SLAM3 fails several times, as indicated by the red stars. These sub trajectories of ORB-SLAM3 are aligned manually for visualization. SVO and DSO fail to track the images.

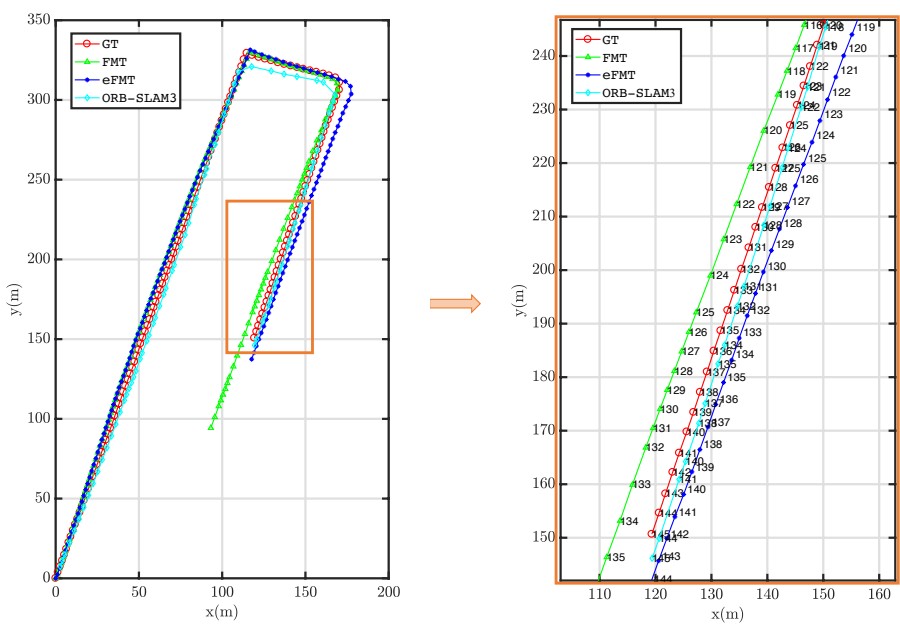

**Figure 13.** Estimated trajectories on the UAV dataset. SVO and DSO fail to track the images.

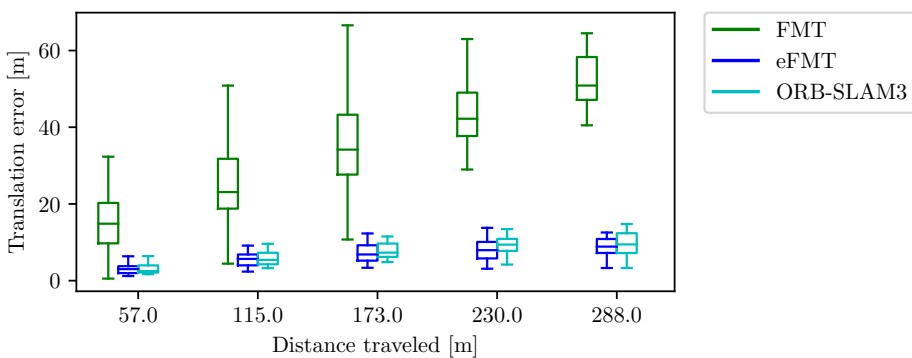

**Figure 14.** Absolute translation error on the UAV dataset. SVO and DSO fail to track the images.

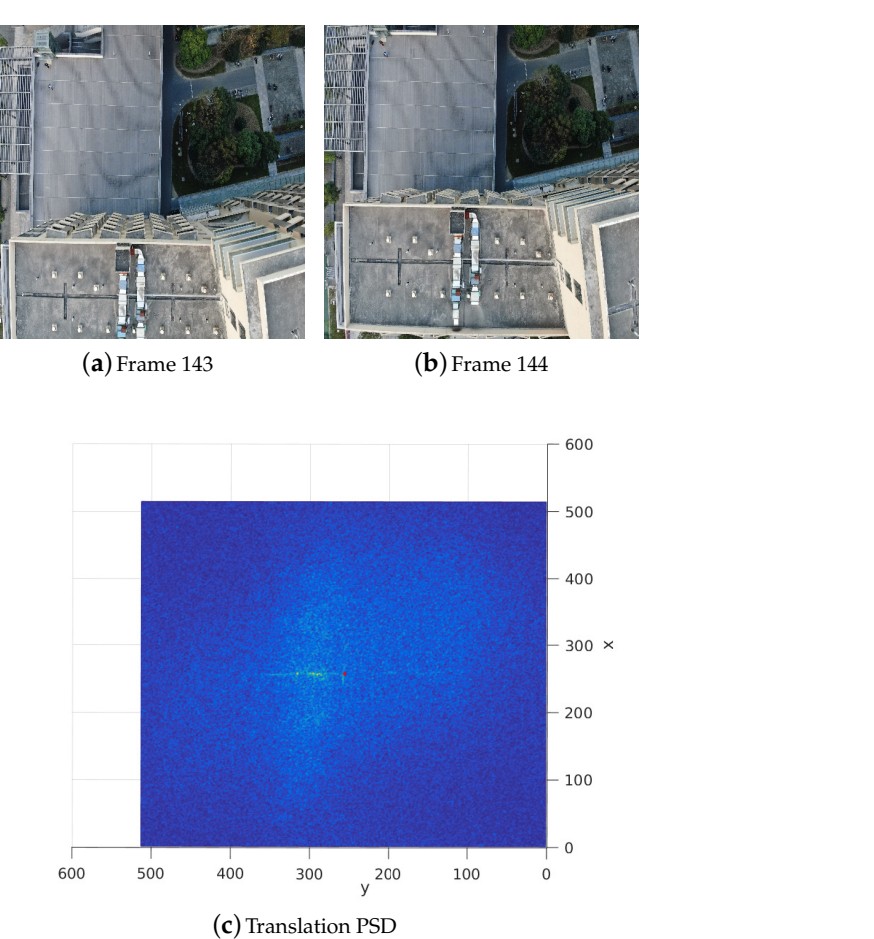

**Figure 15.** Example line segment in the translation phase shift diagram (PSD).

### 6.3. Robustness on Continuous Depth

As observed in our UAV dataset, the buildings in the camera's view may become a slanted plane due to perspective projection. In this case, there will be a continuous line segment in the translation PSD corresponding to the slanted plane (see Figure 15). In this section, we explore the influence of continuous depths on the PSDs, which is important for the performance of eFMT. For that, we simulate a plane and a robot arm in Gazebo, like Figure 8, and then make the robot arm deviate from perpendicular with the plane such that the plane becomes a slanted plane in the camera's view (see Figure 16p). In other words, the depth in the view is now continuous instead of discrete planes, which can be considered to be a limit condition of multi-depth. We collect images by mainly moving the camera along the *y*-axis in the camera's frame (see Figure 8), since it is more complex than

moving along the $x$-axis. When the camera moves along the $x-$axis, the relative depth in the camera view is the same.

　　Figure 16 shows the translation PSDs with different magnitude of the camera's motion. When tilt is $0°$, there is only single depth in the view. It can be found that the darker the blue is, the clearer the high values are. From the second row of Figure 16, we can see that the highest peak distributes to a wide line and the highest energy becomes smaller when the tilt of the camera gets bigger. Since eFMT finds the sector with maximum energy, it can still find the unit translation vector in this condition and implement re-scaling with all the energy in the maximum sector. In contrast, FMT may fail in this case because it only tracks the highest energy. In detail the highest energy peak is prone to change, i.e., prone to be associated with different depths, due to noise and scenario similarity.

　　From each column of Figure 16, it can be seen that the energy distributes more along the $r_{max}$ sector when the camera moves more. This means that it is more obvious that different depths contribute to different pixels. When the motion is small, different depths may contribute to the same pixel due to the image resolution. Please note that there are multiple peaks with high energy, also in the opposite direction, in Figure 16d,e, which is due to the periodic structures in the simulated images. eFMT still has a good chance to work in this case, because the maximum energy sector $r_{max}$ can still reliably be found. Also, the pattern matching will still determine the re-scaling based on the best matching scaling, which should be the correct one, since its energy is highest and fits best.

　　Looking at the different tilt values of Figure 16, we can see that a higher tilt results in a longer line of high energy values and more noise in the PSD. In particular, the PSDs will be too noisy to provide distinguished high values if there is big motion combined with big tilts. For instance, eFMT fails when the tilt is $30°$ or more and the camera motion is bigger than 0.8 m.

　　The rotation and zoom PSDs are shown in Figure 17. Similar to the translation PSDs, more noise will be introduced to the rotation and zoom PSDs with bigger tilt and bigger motion.

　　Overall, multiple depths in the scenario will introduce more noise to the PSDs. When the motion between two frames is not too big, eFMT can still handle the case. However, if the motion is too large, no distinguished energies can be found and eFMT will fail to find the correct sector with maximum energy and thus cannot estimate the motion correctly. Luckily, in the visual odometry task, the motion between two frames is usually not too big. However, it will introduce challenges when we want to do loop closure on the frames with big motion in our future work.

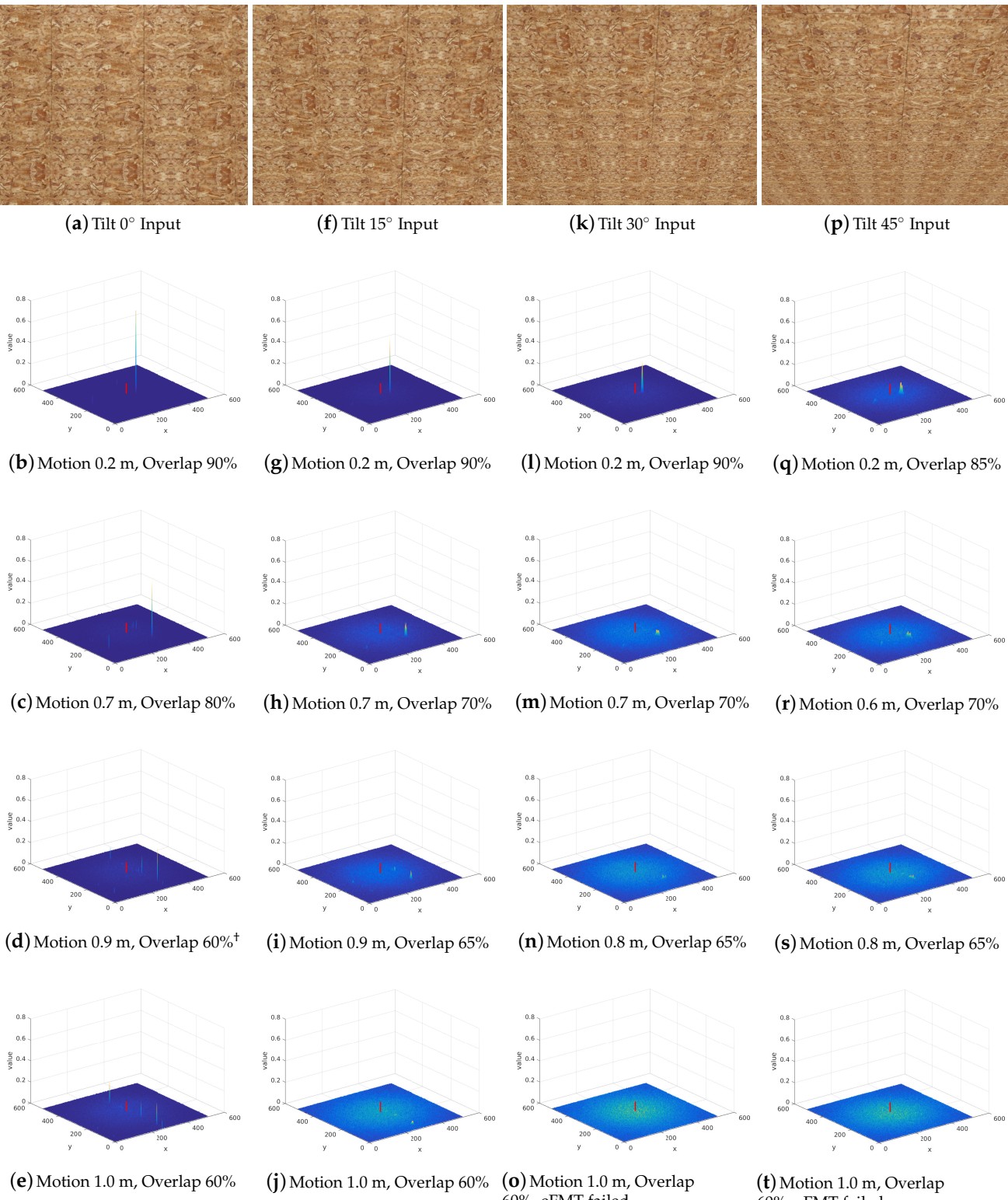

**Figure 16.** Translation PSDs with different tilt angles and camera motions. The angle between the plane and the imaging plane changes from 0° to 45°. Each column corresponds to one angle. When the angle is not 0°, it is a slanted plane in the view, leading to continuous depths. The distance between the camera and the slanted plane is about 3 m. † The overlap is smaller than others in this row due to the bigger motion in *u*-direction of the image.

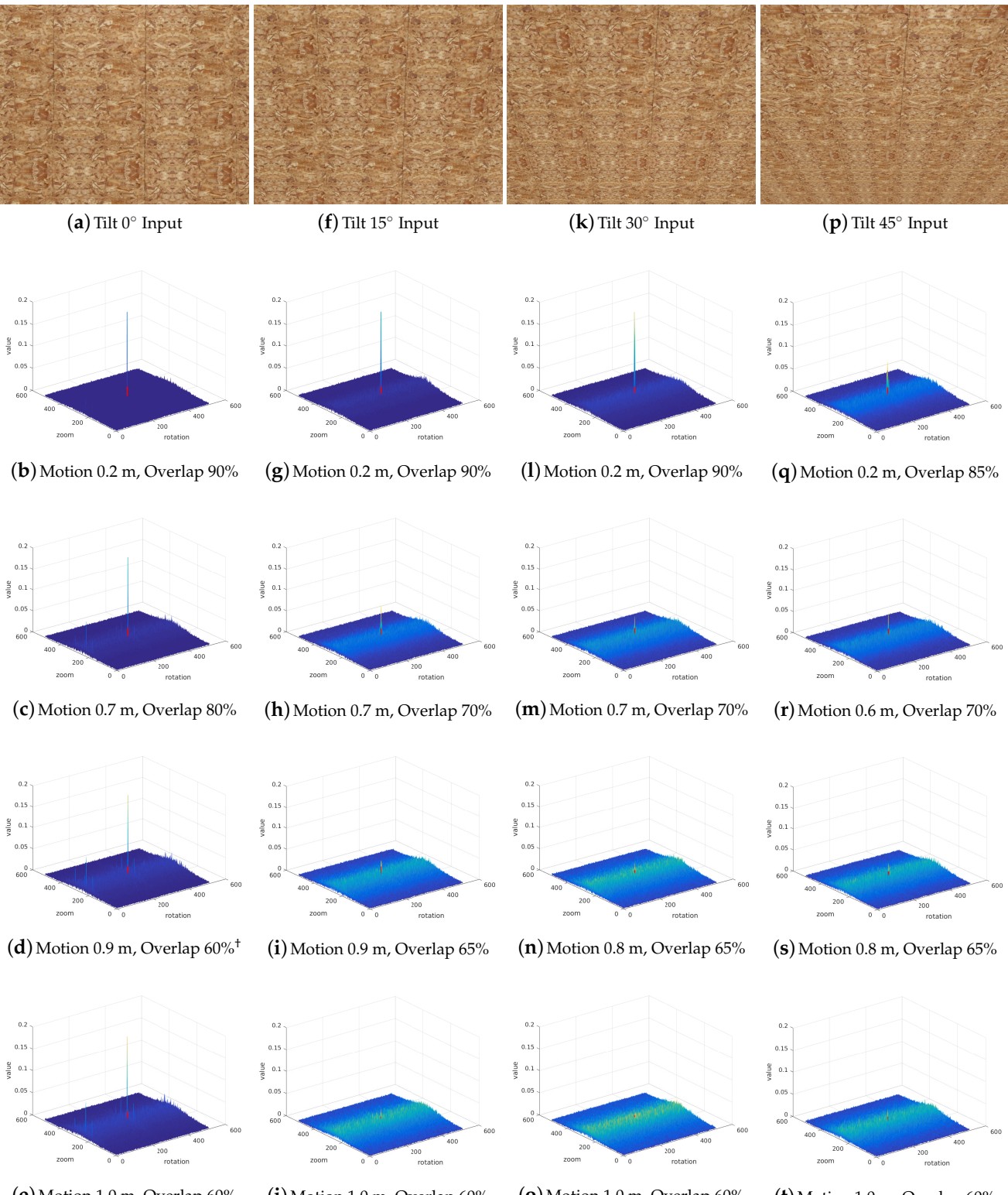

**Figure 17.** Rotation and zoom PSDs with different tilt angles and camera motions. The angle between the plane and the imaging plane changes from 0° to 45°. Each column corresponds to one angle. When the angle is not 0°, it is a slanted plane in the view, leading to continuous depths. The distance between the camera and the slanted plane is about 3 m. † The overlap is smaller than others in this row due to the bigger motion in *u*-direction of the image.

*6.4. Computation Analysis*

Ref. [39] pointed out that image resolution has a big impact on the FMT algorithm and image down-sampling does not hurt FMT performance. In our preliminary test, we find that this still holds for eFMT. The run-time of eFMT is about 0.7 seconds per frame with image resolution $512 \times 512$ and about 0.2 s per frame with a resolution of $256 \times 256$ based on the single-threaded C++ implementation, which is about two times slower than that of FMT. In addition, the multi-translation calculations with multi-zoom values are independent from each other, thus we can compute these in parallel to speed up the algorithm. Thus, eFMT could run as fast as FMT.

## 7. Discussion

In the above experiments, we show that there will be no single energy peak of the PSDs when the scenario changes from single depth to multi-depth. Based on this, eFMT extracts the line with maximum sum energy instead of a single peak, such that it achieves better performance, especially scale consistency, than FMT in multi-depth environments. Our experiments, including one example application on a UAV dataset, show that both FMT and eFMT are more robust than the state-of-the-art VO methods, thanks to the robustness of spectral description against feature-deprived images, while eFMT is successfully removing the constraint for a single-depth scenario that FMT exhibits.

One main limitation of eFMT- and FMT- is that they do not work when there is a roll or pitch in the camera, which is narrowing their scope of application. Smaller deviations from this constraint may be compensated by getting the gravity vector from an IMU and rectifying the images accordingly, but this does not mean that eFMT could be extended to be a general 3D VO algorithm this way.

A major influence on the success of an eFMT image registration is the overlap in the images, which depends on translation amount vs. distance of objects. In the general case, when most of the environment is not tilted more than 45 degree in the frames, our experiments in Section 6.3 indicate that an overlap of 65% or more seems to be sufficient for successful registrations with eFMT. To improve on this, we consider the oversampling strategy in frequency domain proposed in [34], which could make the high energies of PSD more distinguished even with smaller overlap.

In addition, the experiments reveal further problems of eFMT. One is that the accumulated error of eFMT may get bigger when the camera moves for a long time, as shown in Section 6.2.2. This is of course true for all incremental pose estimators. To overcome this, in our future work we will introduce loop closing and pose optimization in the same fashion as current popular VO methods.

Like any monocular VO algorithm, eFMT is up to an unknown scale factor and it will fail if the environment is highly repetitive or does not exhibit enough texture, even though it is better than most other approaches regarding that last constraint.

## 8. Conclusions

This paper extends the classical FMT algorithm to be able to handle zoom and translation in multi-depth scenes. We present a detailed problem formulation and algorithm. Experiments show the clear benefit of our proper re-scaling for Visual Odometry in scenes with more than one depth and compare it to FMT, which indicates that eFMT inherits the advantages of FMT and extends its application scenarios. Moreover, eFMT performs better than the popular VO methods ORB-SLAM3, DSO and SVO in all our experiments, performed on our datasets collected in challenging scenarios.

In our future work, we will continue to make the proposed eFMT more robust and accurate with the following two points. One is to use pose optimization to decrease the accumulated error. Another is exploiting the oversampling strategy to make the PSDs less noisy and the high energies more distinguished.

As we introduced in Section 1, FMT has already been used in all kinds of applications. We can consider the application of eFMT in similar applications if the scenario is multi-

depth, such as underwater robot localization. Underwater turbidity usually prevents feature-based methods to work properly, while spectral methods still have an acceptable performance. Since the bottom of the underwater scenario may be not flat, eFMT is more suitable than FMT.

## 9. Patents

Patent [CN111951318A] from the work reported in this manuscript is under review.

**Supplementary Materials:** The following are available online at https://www.mdpi.com/2072-429 2/13/5/1000/s1.

**Author Contributions:** Conceptualization, S.S. and Q.X.; methodology, S.S. and Q.X.; software, Q.X.; validation, H.K. and Q.X.; formal analysis, Q.X.; investigation, Q.X. and S.S.; resources, Q.X.; data curation, Q.X.; writing—original draft preparation, Q.X.; writing—review and editing, Q.X., S.S. and L.K.; visualization, Q.X.; supervision, S.S.; project administration, S.S.; funding acquisition, S.S. All authors have read and agreed to the published version of the manuscript.

**Funding:** This research was funded by ShanghaiTech grant number 2014F0203-000-11.

**Institutional Review Board Statement:** Not applicable.

**Informed Consent Statement:** Not applicable.

**Data Availability Statement:** The data presented in this study are available in article.

**Conflicts of Interest:** The authors declare no conflict of interest.

## Abbreviations

The following abbreviations are used in this manuscript:

| | |
|---|---|
| VO | Visual odometry |
| ORB-SLAM | ORB based simultaneous localization and mapping |
| DSO | Direct sparse odometry |
| SVO | Semi-direct visual odometry |
| FMT | Fourier-Mellin transform |
| eFMT | Extended Fourier-Mellin transform |
| PSD | Phase shift diagram |
| RTK | Real-time kinematic |
| GPS | Global positioning system |
| SNR | Signal-to-Noise ratio |

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
