# Peer review of "Rethinking the Fourier-Mellin Transform: Multiple Depths in the Camera’s View"

_remotesensing, doi:10.3390/rs13051000_

Round 1
Reviewer 1 Report
This paper presents a logical method to improve FMT-based registration and substantiates it with intelligent and novel algorithm design choices based on the particular nature of the the PSD. However, the paper does require a lot of faith from the reader about the claims underpinning some of those choices.
In particular, it stands to reason that there are some scenarios in which the eFMT method may fail, e.g. when there could be scale confusion caused by periodic structures in the image, or when the majority of the scene cannot clearly be defined in discrete height planes. The paper has not convinced me that a significant deviation of the camera orientation from perpendicular with the plane would not pollute the PSDs to the point that peak finding fails. Not all of these scenarios have to be covered; FMT itself was also limited in application scope and this paper extends the scope. However, the new scope of application is poorly delineated in my opinion, perhaps by a reluctance of the authors to stress their own method to failure point in synthetic experiments.
On the topic of those experiments, the inclusion of simulations should serve to easily generate large numbers of tests, whereas in this work the artificial example is very basic, or perhaps only this basic example is shown. As I requested above, it would be nice to explore the limits of robustness more in simulation and present this to the reader.
Overall, I think the method itself is novel enough to warrant publication, and while I did not have time to verify the mathematics in detail, the main derivation seems to hold true. Where the paper could be significantly improved however, is by formulating stronger conclusions supported by more evaluation.
Below follow some more detailed comments about the paper.
- It is unclear how the minimum and maximum zoom should be determined from the column in eq.18, and how this limits application scenarios.
- Diagram in Fig. 4 could be clearer with respect to where rotation information is actually computed (it now appears to be calculated from the "zoom energy vector" but to my understanding it is extracted simultaneously with the zoom vector).
- Although Section 4.4 addresses the main concern about pre-zooming for translation calculation, it does not give a clear picture of how much the noise introduced by deviating depth regions will spread out relative to the difference in depth. The section suggests that the concentration of evidence disperses very quickly even from minor depth differences, but I am not entirely convinced this is the case.
- Algorithm 2 seems a very basic implementation that is doing an unnecessary amount of iterations. The motivation for such a brute force approach is poor; is the computation of the scale so prone to local maxima that a gradient descent type of approach does not work?
- Table 1 has a small typo (third measurement presumably should be z20 not z10).
- The end of Section 6.2 mentions error accumulation as a weakness of the method, but this weakness is not exclusive to the method and applies to any incremental pose estimator.
- The mention of fusion with IMU in the conclusion is very superficial, the inclusion of inertial measurements in a VO framework is far from straightforward and this section is very shallow in general. In my opinion it would be better to identify weaknesses in the method and address those in future work planning.
Reviewer 2 Report
Authors present an interesting and sound work for VO with image processing based on different scale treatment and FMT. Despite the soundness and validity, my main concern falls on the sort of dataset/benchmark (assumed to be custom). Apart from simulation experiments, and comparison against the other state-of-the-art methods, results should be also tested against some publicly available datasets. Please take that thoroughly into account.
As a minor comment:
I miss some more discussion on similar holistic descriptors that compute images globally as a whole appearance data. I.e., authors like L. Payá, have conducted interesting research on how to estimate (amongst others) height from Fourier transform, Gabor, HOG, etc. Please consider similar works for initial discussion at the Intro
https://orcid.org/0000-0002-3045-4316
Round 2
Reviewer 2 Report
I congratulate authors for their effort in replying my comment. Now I clearly understand why the did not performed with real dataset. I do not find necessary to include the figure, but certain explanation/justification for this decision, as it has been given to me (briefly, as already initiated in the manuscript in the Intro, but also with a slight mention in the results section).
